# Jointly Reinforcing Diversity and Quality in Language Model Generations

## Abstract

Post-training of Language Models (LMs) often prioritizes accuracy and helpfulness at the expense of diversity. This creates a tension: while post-training improves response quality, it also sharpens output distributions and reduces the range of ideas, limiting the usefulness of LMs in creative and exploratory tasks such as brainstorming, storytelling, or problem solving. We address this challenge with Diversity-Aware Reinforcement Learning (Darling), a framework that jointly optimizes for response quality and semantic diversity. At its core, Darling introduces a learned partition function to measure diversity beyond surface-level lexical variations. This diversity signal is then combined with a quality reward during online reinforcement learning, encouraging models to generate outputs that are both high-quality and distinct. Experiments across multiple model families and sizes show that Darling generalizes to two regimes: non-verifiable tasks (instruction following and creative writing) and verifiable tasks (competition math). In the first setting, Darling consistently outperforms quality-only RL baselines on 5 benchmarks, producing outputs that are simultaneously of higher quality and novelty. In the second setting, it achieves both higher pass@1 (quality) and pass@k (diversity). Most strikingly, explicitly optimizing for diversity catalyzes exploration in online RL, which manifests itself as higher-quality responses.

## 1 Introduction

Diversity—the ability to generate varied ideas, outputs, or solutions—plays a critical role in driving innovation across real-world applications (Guilford, 1950; Lu et al., 2025). For example, in scientific discovery, diverse hypotheses or experimental outcomes enable researchers to explore a broader solution space, potentially uncovering novel insights and breakthroughs (Gruver et al., 2023; Romera-Paredes et al., 2024; Si et al., 2025). Similarly, in other tasks such as creative writing (Fan et al., 2018) and natural conversations (Li et al., 2016a), diverse outputs are essential for innovation that requires avoiding repetitive or predictable outcomes. In reinforcement learning (RL) and self-training loops of LMs, diversity is also crucial. Policies that produce diverse outputs enable thorough exploration of the action space, critical for discovering novel and effective strategies (Chen et al., 2025a; Cheng et al., 2025; Wu et al., 2025a; An et al., 2025).

However, recent developments in Language Models (LMs) have revealed a significant issue: post-training of LMs often results in overly sharpened output distributions (Huang et al., 2025; Li et al., 2025b), leading to a significant reduction of diversity among generated responses (Padmakumar & He, 2024; Shypula et al., 2025), even sharing identical prefixes (Ji et al., 2025) or becoming near duplicates (Mahony et al., 2024; Zhang et al., 2024), reducing the overall informativeness of outputs (Lin et al., 2021; Kirk et al., 2024; West & Potts, 2025; Yang & Holtzman, 2025; Yun et al., 2025).

To address the loss of diversity during LM post-training, we propose Diversity-Aware Reinforcement Learning (Darling), an online RL objective that (a) measures diversity at the *semantic* level via a learned classifier, and (b) fuses diversity and quality to condition gradient updates on "usefully different" trajectories. As illustrated in Figure 1, Darling first partitions rollouts from a single user prompt into distinct clusters using a semantic classifier, capturing diversity beyond superficial lexical differences (§3.1). It then combines (multiplies) the diversity assessment with a quality reward, amplifying the advantage of log-probabilities for responses that are both high-quality and

Figure 1: Diversity-Aware Reinforcement Learning (DARLING): We first partition LLM generations into semantically equivalent clusters (represented by colors). While standard GRPO (Shao et al., 2024) increases probabilities based on response quality only, DARLING amplifies the increase in probability of diverse and high-quality responses. High quality responses get positive advantage whereas low quality responses get negative advantage.

semantically diverse in online RL (§3.2). Unlike standard GRPO, which pays no heed to diversity, DARLING explicitly encourages exploration across meaningfully distinct response clusters.

We validate DARLING's effectiveness and generalizability across both non-verifiable and verifiable tasks, over different language model families and sizes. Experimental results demonstrate that DARLING preserves the original model's diversity and achieves improved benchmark performance in both non-verifiable instruction-following and creative writing tasks, and verifiable math problems.

In summary, our contributions are three-fold:

(1) We propose DARLING, an RL framework that simultaneously optimizes quality and diversity, preventing diversity collapse during post-training.

(2) We demonstrate that a learned semantic classifier can serve as a scalable and generalizable signal of diversity to integrate into online RL training.

(3) We validate that explicitly optimizing for diversity in online RL for LMs promotes greater exploration, often leading to improvements in quality in both non-verifiable (creative writing) and verifiable (competition math) benchmarks.

## 2 NOTATIONS AND PRELIMINARIES

Let $\mathcal{S}$ denote the set of natural language token sequences, a language model $\pi(\cdot \mid x)$ takes a token sequence $x \in \mathcal{S}$ as its input and outputs a probability distribution over $\mathcal{S}$. We denote the probability of a specific token sequence $y \in \mathcal{S}$ as $\pi(y \mid x)$ and denote the token at position $t$ as $y^t$. Given a reward function $r : \mathcal{S} \times \mathcal{S} \to \mathbb{R}$ which maps a pair of natural language instructions and responses $x, y \in \mathcal{S}$ to a scalar value $r(x, y) \in \mathbb{R}$, LM post-training aims to solve the following KL constrained optimization problem:

$$\max_{\pi} \mathbb{E}_{x \sim \mathcal{D}, y \sim \pi(\cdot|x)} \left[ r(x, y) - \beta \frac{\pi(y \mid x)}{\pi_{\text{ref}}(y \mid x)} \right] = \max_{\pi} \mathbb{E}_{x \sim \mathcal{D}, y \sim \pi(\cdot|x)} [r(x, y)] - \beta \mathbb{D}_{\text{KL}} (\pi || \pi_{\text{ref}}), \quad (1)$$

where $\mathcal{D}$ is a dataset of prompts and $\pi_{\text{ref}}$ is a reference model from which we do not want to deviate too much, usually implemented as the LM before the optimization process. We will omit the KL divergence term for simplicity. Group Relative Policy Optimization (GRPO; Shao et al. (2024)) optimizes (1) by maximizing the following objective:

$$\mathbb{E}_{x \sim \mathcal{D}, \{y_i\}_{i=1}^n \sim \pi_{\text{act}}(\cdot|x)} \left[ \frac{1}{n} \sum_{i=1}^n \frac{1}{|y_i|} \sum_{t=1}^{|y_i|} \left( \min \left( \text{IS}_{i,t} \cdot A_{i,t}, \ \text{clip}(\text{IS}_{i,t}, 1 - \varepsilon, 1 + \varepsilon) \cdot A_{i,t} \right) \right) \right] \quad (2)$$

where $n$ is the number of responses per prompt, and

$$\text{IS}_{i,t} = \frac{\pi_\theta(y_i^t \mid y_i^{<t}, x_i)}{\pi_{\text{act}}(y_i^t \mid y_i^{<t}, x_i)}$$

is the importance sampling (Kloek & van Dijk, 1978) ratio between the current policy $\pi_\theta$ and the actor $\pi_{\text{act}}$ (the model used to generate $y_i$), and

$$A_{i,t} = \frac{r(x, y_i) - \text{mean}_{j=1}^n(r(x, y_j))}{\text{std}_{j=1}^n(r(x, y_j))} \tag{3}$$

is the advantage of the response $y_i$, measuring how much better (or worse) $y_i$ is over an average response, and $\varepsilon$ is a hyperparameter preventing the importance sampling term $\text{IS}_{i,t}$ from being too large or small. GRPO and its variants (Yu et al., 2025; Liu et al., 2025c; Hu, 2025) are widely adopted as some of the go-to algorithms for LM post-training (DeepSeek-AI et al., 2025; Liu et al., 2025a; Yang et al., 2025a) due to their simplicity and stability.

## 3 METHOD: DARLING

Figure 1 illustrates DARLING (Diversity Aware Reinforcement Learning). We first partition the responses using our developed classifier (§3.1) that captures semantic similarity, then combine diversity and quality signals in an RL framework to generate diverse and high-quality responses (§3.2).

### 3.1 PARTITIONING THE RESPONSES INTO SEMANTIC EQUIVALENCE CLASSES

We begin by formally defining *diversity*, as used in our work: Given a pairwise distance metric $d : \mathcal{S} \times \mathcal{S} \rightarrow \mathbb{R}^+$ between two generations, and a group of $n$ generations $y_1, \cdots, y_n$ we define the diversity of a generation $y_i$ with respect to all other generations $y_j$ ($j \neq i$) as the average pairwise distance between $y_i$ and all other generations $y_j$ ($j \neq i$):

$$\text{Div}_d(y_i \mid y_1, \cdots, y_n) = \frac{\sum_{j=1, j \neq i}^n d(y_i, y_j)}{n-1}. \tag{4}$$

We aim to incorporate a scalable metric of semantic diversity that captures deeper differences beyond surface-level variation into our training process. Following Zhang et al. (2025b), we train a binary classifier to determine whether two responses convey equivalent semantics:

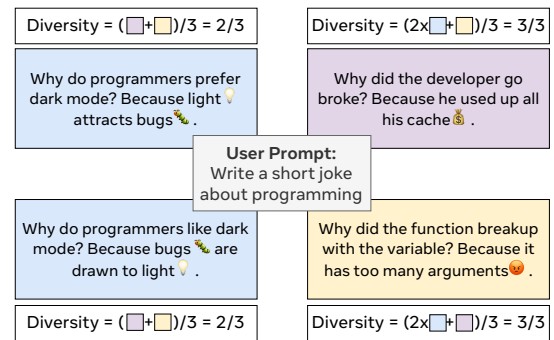

Figure 2: Example of partitioning a group of responses into semantically equivalent subgroups and evaluating diversity. *Diversity* is calculated as the normalized count of responses that is distinct from a given response.

$$\textbf{classify}(y_i, y_j) = \mathbf{1}(y_i \text{ semantically equivalent to } y_j).$$

Details on how we trained this classifier can be found at Appendix A. Responses predicted as equivalent are clustered to form a partition of all responses into semantic clusters, where multiple members of the same cluster provide little additional value beyond a single representative. This approach parallels semantic entropy methods (Kuhn et al., 2023), which group semantically equivalent responses.

We directly set diversity metric $d = \textbf{classify}(\cdot, \cdot)$. Figure 2 illustrates an example of our diversity calculation from partitions. For a single prompt *"Write a short joke about programming."*, responses in the left column (blue) are classified as semantically equivalent, both utilizing that the word "bug" has multiple meanings. The responses in the right column (purple and yellow) are distinct from the three other responses. For each of the individual responses in blue boxes, there are only two other responses that are distinct: purple and yellow. Therefore, using (4), we derive the diversity of both blue responses as 2/3. Similarly, the yellow and purple responses have diversity 3/3 because they are distinct from all other responses.

## 3.2 DARLING: DIVERSITY AWARE REINFORCEMENT LEARNING

Given a diversity function $\text{Div}_d$ and a reward function $r$, we define the diversity-aware reward as:

$$r_{\text{darling}}(x, y_i \mid y_1, \cdots, y_n) := r(x, y_i) \times \text{Norm}\big(\text{Div}_d(y_i \mid y_1, \cdots, y_n)\big), \tag{5}$$

where $x$ is the input prompt and $y$ is the model response. $\text{Norm}(\cdot)$ normalizes diversity values to be between 0 and 1.

We choose to multiply the two reward scores instead of adding them. While simply adding the quality and diversity rewards is an alternative approach, this method poses challenges due to the differing scales of the two rewards. Naively summing the two rewards can lead the model to prioritize one reward over the other. Ablation studies of varying methods of fusing reward scores are in §6.1.

DARLING plugs (5) into (1), which amplifies the effective reward $r_{\text{darling}}$ of high reward responses that are diverse from others. Motivated by prior work (Liu et al., 2025c; Yu et al., 2025), we also make the following modifications: changing sequence-level loss averaging to token-level averaging in (2) as the former exhibits bias towards longer sequences, and removing normalization by standard deviation in (3) since it amplifies the noise in dense rewards. We leave detailed ablations on the effect of advantage normalization to Appendix B.

The overall loss function for DARLING is thus defined as:

$$\mathbb{E}_{x \sim \mathcal{D}, \{y_i\}_{i=1}^n \sim \pi_{\text{act}}(\cdot \mid x)} \left[ \frac{1}{\sum_{i=1}^n |y_i|} \sum_{t=1}^{|y_i|} \Big( \min \big(\text{IS}_{i,t} \cdot A_{i,t}^{\text{darling}}, \; \text{clip}(\text{IS}_{i,t}, 1 - \varepsilon, 1 + \varepsilon) \cdot A_{i,t}^{\text{darling}}\big) \Big) \right],$$
$$\tag{6}$$

where we use the diversity-aware reward $r_{\text{darling}}$ as the effective reward:

$$A_{i,t}^{\text{darling}} = r_{\text{darling}}(x, y_i \mid y_1, \cdots, y_n) - \text{mean}_{j=1}^n \big(r_{\text{darling}}(x, y_j \mid y_1, \cdots, y_n)\big).$$

Compared to standard GRPO, our main modification is that we multiply a normalized diversity reward $\text{Norm}\big(\text{Div}_d(y_i \mid y_1, \cdots, y_n)\big)$ by the quality reward $r(x, y)$ to promote high-quality and diverse rewards during training. This amplifies the increase in the log-likelihood of responses that are both high quality and diverse — jointly reinforcing diversity and quality.

## 4 DARLING ON NON-VERIFIABLE TASKS

We first show the effectiveness of DARLING on general non-verifiable instruction-following tasks. We describe our setup in §4.1. We show and analyze our results in §4.2.

### 4.1 SETUP

**Models and Baselines** We use two models of different scales: `Llama-3.1-8B-Instruct` and `Llama-3.3-70B-Instruct` (Llama Team, 2024) as reference models and perform training on a randomly sampled subset of 10k prompts from WildChat (Zhao et al., 2024), the same setup as in Lanchantin et al. (2025a). We use `Nexusflow/Athene-RM-8B` (Frick et al., 2024) as the reward function in (5) for quality. We use a batch size of {32 (8B), 64 (70B)} prompts and 8 rollouts per prompt. Other training hyperparameters can be found in Appendix C.

We compare our method against the following baselines:

- GRPO (Shao et al., 2024): The method described in §2, with token-level mean aggregation;
- DivPO (Lanchantin et al., 2025a): a DPO-based (Rafailov et al., 2024) optimization method that selects the *most* diverse response among the high-quality ones as the chosen response and the *least* diverse response as the rejected response;
- GRPO-Unlikeliness (He et al., 2025a): a revised GRPO algorithm that assigns higher rewards to low-likelihood responses to encourage greater diversity.

We implement DARLING using the verl codebase (Sheng et al., 2024). The original classifier of Zhang et al. (2025b) was limited to a context length of 512 tokens[1]. In our work, we extend their

---

[1] https://github.com/novelty-bench/novelty-bench/blob/main/src/partition.py#L69

method by training a classifier with an 8192-token context window, using the same human-annotated data. Details of this training procedure are in Appendix A.1.

**Evaluation Benchmarks and Metrics** For evaluating response quality, we employ standard benchmarks: AlpacaEval 2.0 (Li et al., 2023; Dubois et al., 2024), ArenaHard v0.1/2.0 (Li et al., 2025a), and EQ-Bench (Creative Writing) (Paech, 2023). We report the length-controlled win rate (LCWR) for AlpacaEval 2.0 and the win rate with style control (markdown, length) for ArenaHard on the creative writing prompts. We report the normalized ELO score for EQ-Bench. For both AlpacaEval and ArenaHard, we follow Lanchantin et al. (2025b) and use GPT-4o (OpenAI et al., 2024) as the judge. For EQ-bench, we use Claude 3.7 Sonnet (Anthropic, 2024) as the judge. To evaluate diversity, we use NoveltyBench (Zhang et al., 2025b) and report the number of *semantically* distinct generations (Dist) and the average number of distinct 4grams (Dist-4) normalized by sequence length. We provide detailed descriptions of the benchmarks in Appendix E.

## 4.2 EXPERIMENTAL RESULTS

| | AE 2.0* | AH v2.0* | AH v0.1* | EQ-Bench | NoveltyBench | |
|---|---|---|---|---|---|---|
| | LCWR (%) | WR (%) | WR (%) | ELO | Dist (#) | Dist-4 (%) |
| Llama-3.1-8B-Inst | 31.9 | 7.1 | 30.9 | 636 | 5.28ch | 93.9 |
| GRPO | 48.7 | 61.1 | 45.5 | 659 | 2.08 | 92.8 |
| DivPO | 43.5 | 54.4 | 39.7 | 639 | 4.34 | 94.1 |
| GRPO-Unlikeliness | 45.6 | 59.5 | 46.2 | 724 | 3.53 | 93.2 |
| DARLING | **55.2** | **68.8** | **63.7** | **905** | **5.49** | **96.0** |
| Llama-3.3-70B-Inst | 44.6 | 17.7 | 64.9 | 737 | 2.95 | 91.7 |
| GRPO | 73.3 | 89.7 | 79.2 | 1261 | 2.31 | 94.6 |
| DivPO | 58.0 | 59.1 | 72.4 | 1041 | 3.67 | 94.9 |
| GRPO-Unlikeliness | 69.5 | 84.2 | 76.4 | 1346 | 3.15 | 95.2 |
| DARLING | **80.4** | **91.2** | **85.7** | **1531** | **4.26** | **95.3** |

Table 1: Non-verifiable Task Evaluations. For each method, we train a single model on 10,000 WildChat prompts. We evaluate the models on AE (AlpacaEval 2.0 Length-Controlled Win Rate), AH v2.0/v0.1 (ArenaHard, creative writing subset), EQ-Bench (ELO), and NoveltyBench. * indicates we used GPT-4o as the judge. All metrics are the higher the better. We find that models trained with DARLING achieve the best quality measured by both AlpacaEval/ArenaHard win rates and EQ-Bench ELO, and simultaneously are the most diverse, as measured by NoveltyBench.

**DARLING achieves both the best quality and diversity across all benchmarks.** Table 1 shows our main results: we observe that DARLING is able to jointly optimize for both quality and diversity. Specifically, DARLING results in the best quality scores (AlpacaEval and ArenaHard win rates) across our baselines, while also achieving the best diversity in both semantic level (Distinct) and lexical level (Distinct-4), showcasing the effectiveness of our method. Moreover, although we did not explicitly train on creative writing prompts, DARLING achieves the best ELO score compared to all baselines in EQ-Bench (creative writing), demonstrating the effectiveness of improving diversity on creative tasks. We show generation examples and perform qualitative analysis of models trained with DARLING in Appendix D.

**DARLING improves the Pareto front between quality and diversity by varying sampling temperature.** We further investigate the effect of sampling temperature on the quality-diversity Pareto front. We vary the sampling temperature ($T = \{0.2, 0.4, 0.6, 0.8, 1.2\}$) of two models (Llama-3.1-8B-Instruct and Llama-3.3-70B-Instruct), being trained with GRPO and DARLING. Figure 3 shows that DARLING (blue) exhibits both better quality and better diversity than both the baseline (green) and GRPO (orange) at both scales, pushing forward the pareto-front of the "quality-diversity tradeoff" (Zhang et al., 2021; Padmakumar et al., 2025).

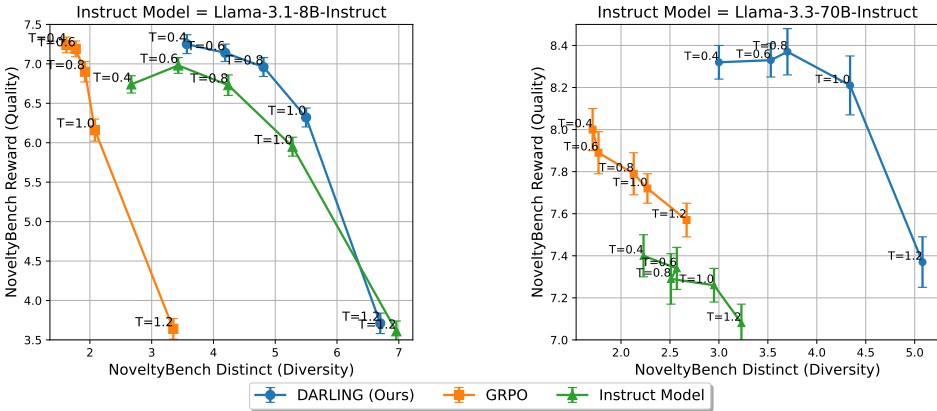

Figure 3: The quality-diversity tradeoff when using different sampling temperatures ($T$) for models (at 8B and 70B scales) trained with standard GRPO and DARLING. $X$-axis: Distinct metric in NoveltyBench; $Y$-axis: Reward score used in NoveltyBench measuring quality of responses. DARLING (blue) simultaneously achieves better quality (y-axis) and diversity (x-axis) as demonstrated by the improved Pareto fronts on both the 8B and 70B scale.

## 5 DARLING ON VERIFIABLE TASKS

In this section we present experimental results of DARLING on verifiable math problems. We describe our setup in §5.1 and show our results in §5.2.

### 5.1 SETUP

**Models, Baselines, and Benchmarks** We train models on top of `Qwen3-4B-Base` and `Qwen3-14B-Base` (Yang et al., 2025a) using the DeepscaleR dataset (Luo et al., 2025), where we first filter out questions that are unanswerable due to missing figures, and then subsample 10,000 examples. We compare our method against GRPO (Shao et al., 2024) on four competition math benchmarks: AIME25 (Art of Problem Solving, 2025), OlympiadBench (He et al., 2024), HMMT 2025, and Brumo 2025 (Balunović et al., 2025). We deliberately choose these benchmarks because they were released concurrently with `Qwen3`, preventing the effect of potential data contamination. We use the Hugging Face `Math-Verify`[2] library to automatically check the correctness of model answers, assigning a binary reward of $r = 1$ for correct and $r = 0$ for incorrect solutions. Following prior work (Yue et al., 2025), we report pass@1 scores as a measure of quality and pass@k scores as a measure of diversity. To evaluate pass@k up to $k = 128$, we sample $n = 256$ responses for each prompt, and we average the performance of the 256 examples for calculating pass@1 to account for the variance introduced by the relatively small sizes of these benchmarks. We use the method in Chen et al. (2021) for an unbiased estimate of pass@k from $n = 256$ examples:

$$\text{pass@}k := \mathbb{E}\left[1 - \frac{\binom{n-c}{k}}{\binom{n}{k}}\right], \tag{7}$$

where $c$ is the number of correct generations. With increased diversity, we expect to see an improved performance of pass@k as the model is more likely to hit the correct answer when it generates more diverse responses. Additional hyperparameters can be found at Appendix C.

**Training an equivalence classifier for math** For building the diversity classifier, Zhang et al. (2025b) collected annotated training data which are sampled prompts from WildChat (Zhao et al., 2024) filtered for non-verifiable tasks. To adapt their method to math, we sample prompts from DeepscaleR and collect trajectories generated by 8 different models spanning multiple model families and sizes. We annotate whether a pair of trajectories is semantically equivalent using `Llama-3.3-70B-Instruct` (Llama Team, 2024). We then finetune a

---

[2]https://github.com/huggingface/Math-Verify

`Qwen3-Embedding-4B` (Zhang et al., 2025a) model using the annotations to obtain our math semantic equivalence classifier. Details on how we perform trajectory sampling, annotations, and classifier training can be found in Appendix A.2.

## 5.2 EXPERIMENTAL RESULTS

**DARLING improves both pass@1 and pass@k in competition math.** Figure 4 shows our main results: we plot pass@k from k=1 to k=128. We observe that DARLING outperforms the GRPO baseline in both quality and diversity. First, for pass@1 (as a measure of quality), DARLING outperforms GRPO by +3.51/1.90% averaged across 4 benchmarks for 4B and 14B models respectively. Next, for pass@128 (as a measure of diversity), DARLING outperforms GRPO by +7.62/10.16%. This shows that by jointly reinforcing quality and diversity, DARLING is able to achieve the best of both worlds in competition math benchmarks — simultaneously achieving better pass@1 and pass@k. Furthermore, we observe the largest gains on HMMT, the most challenging of the four datasets, suggesting that enhanced exploration yields greater improvements on harder datasets. We report accuracy for each of the datasets in Appendix F.

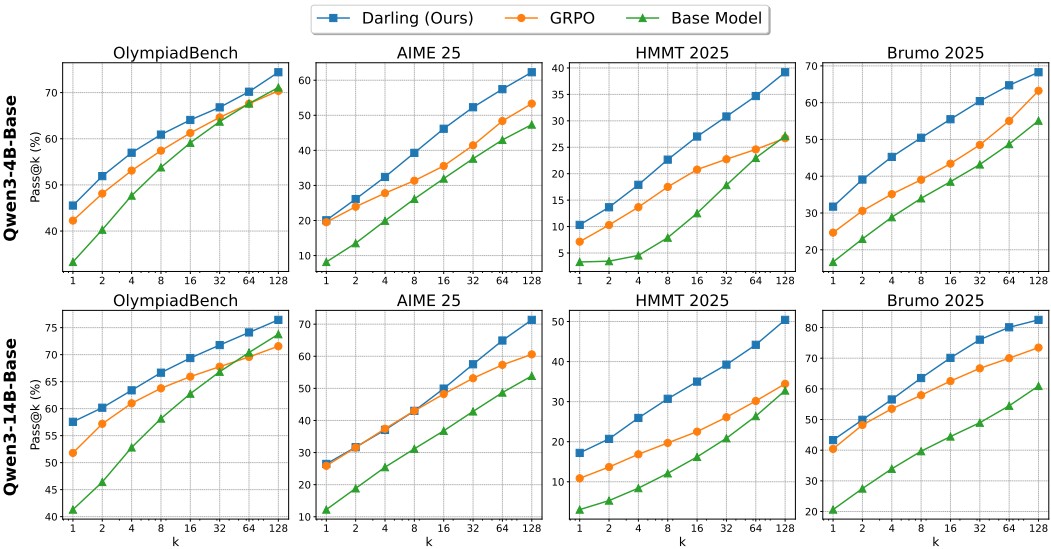

Figure 4: Comparison of different pass@k values by applying GRPO and DARLING on `Qwen3-4B-Base` and `Qwen3-14B-Base` on competition math benchmarks. DARLING outperforms GRPO simultaneously for pass@1 (+3.51/+1.9% on avg.) and pass@128 (+7.62/10.16% on avg.) on 4B and 14B models respectively. DARLING simultaneously achieves the best quality and diversity averaged across 4 competition math benchmarks.

## 6 ABLATIONS

In this section, we perform ablations on the design choices of DARLING. In particular, we compare additive aggregation of quality and diversity rewards v.s. multiplicative aggregation (ours) in §6.1 and we compare our partition classifier that measures semantic diversity with traditional lexical diversity in §6.2. We defer the ablation of advantage normalization to Appendix B.

### 6.1 ABLATIONS ON MULTIPLICATIVE V.S. ADDITIVE AGGREGATION

A naive aggregation of the quality reward $r$ and the diversity signal $\text{Div}_d$ is to add the two rewards, as many existing works perform *additive* aggregation of the quality-based reward and auxiliary rewards such as length (Aggarwal & Welleck, 2025; Liu et al., 2025b), entropy (Cheng et al., 2025), and format (Wu et al., 2025b).

We ablate the effect of adding or multiplying the quality reward $r$ and the diversity reward Norm(Div) in equation 5 in our non-verifiable setting and report the results in Table 2. To ag-

gregate the two rewards, we manually normalize (subtract mean and divide by standard deviation) both $r$ (the quality reward) and $\text{Div}_d$.

| | AlpacaEval 2.0 | ArenaHard v2.0 | | NoveltyBench |
| --- | --- | --- | --- | --- |
| | LCWR(%) | creative writing | hard prompts | Distinct (#) |
| Baseline | 31.92 | 7.1 (-1.4 / +1.8) | 6.4 (-0.8 / +1.3) | 5.28 |
| Quality (GRPO) | 48.74 | 61.1 (-3.5 / +4.5) | **33.9** (-2.3 / +2.5) | 2.08 |
| Quality + partition | 53.17 | **69.2** (-3.6 / +3.9) | 32.7 (-3.2 / +2.9) | 5.23 |
| DARLING = Quality × partition | **55.15** | 68.8 (-3.3 / +2.9) | 31.1 (-2.0 / +2.1) | **5.49** |

Table 2: Ablation comparing the way we aggregate the quality and diversity reward: additive v.s. multiplicative in Equation 5, evaluating on AlpacaEval 2.0, ArenaHard v2.0 (with Style Control), and NoveltyBench. Baseline = `Llama-3.1-8B-Instruct`.

We observe that multiplicative aggregation (DARLING) outperforms additive aggregation in AlpacaEval 2.0, and performs similarly in ArenaHard v2.0 and NoveltyBench. We opt for multiplicative aggregation due to its simplicity as it does not require additional handling of mismatched reward scales and hyperparameter tuning of mixing weights on individual rewards.

## 6.2 Ablations on Lexical Metrics for Diversity

We study whether our proposed partition classifier can be replaced by a simple lexical diversity metric. We replace our semantic equivalence classifier with the number of distinct $N$-grams in online RL training. Specifically, we set the diversity of a response $y_i$ w.r.t all other responses for the same input prompt as:

$$\text{Div}_{\text{ngram}}(y_i \mid y_1, \cdots, y_n) = \frac{\text{number of distinct } N\text{-grams that only appear in } y_i}{\text{total number of } N\text{-grams in } y_i}.$$

This means that if no $N$-gram in $y_i$ appears in any other response $y_j (j \neq i)$, then the diversity $\text{Div}_{N\text{-gram}}(y_i \mid y_1, \cdots, y_n) = 1$. Similarly, $\text{Div}_{N\text{-gram}}(y_i \mid y_1, \cdots, y_n) = 0$ if all $N$-grams in $y_i$ appear in at least one other response. In our experiments, we set $N = 4$ and denote this setting as "Quality × 4gram". We report experimental results on non-verifiable tasks in Table 3.

| | AlpacaEval 2.0 | ArenaHard v2.0 | | NoveltyBench |
| --- | --- | --- | --- | --- |
| | LCWR(%) | creative writing | hard prompts | Distinct (#) |
| Baseline | 31.92 | 7.1 (-1.4 / +1.8) | 6.4 (-0.8 / +1.3) | 5.28 |
| Quality (GRPO) | 48.74 | 61.1 (-3.5 / +4.5) | **33.9** (-2.3 / +2.5) | 2.08 |
| Quality × 4gram | 53.82 | **71.9** (-3.3 / +3.6) | 31.3 (-2.3 / +2.9) | 3.59 |
| DARLING = Quality × partition | **55.15** | 68.8 (-3.3 / +2.9) | 31.1 (-2.0 / +2.1) | **5.49** |

Table 3: Comparison of $N$-gram diversity loss to DARLING. The $N$-gram diversity loss (N=4) performs similarly to DARLING in terms of quality, but underperforms DARLING in terms of diversity in NoveltyBench. Baseline = `Llama-3.1-8B-Instruct`.

We observe that while 4gram diversity integrated with quality is able to match the performance of DARLING in LM-as-a-Judge evaluations (AlpacaEval 2.0, ArenaHard v2.0), it significantly underperforms DARLING in semantic diversity assessment (NoveltyBench).

Additionally, we evaluate the performance of using 4gram diversity in competition math and report the results in Table 13. We found that in math questions, using lexical diversity as a reward underperforms the GRPO baseline in terms of pass@1 performance. We analyze the the generations and observe that the policy often hacks the ngram diversity reward by generating texts that are of a different language, or self-reflections about the difficulty of the problem. We provide an example of such ngram reward hacking in Appendix G.

| Model | Pass@128 | | | | Pass@1 | | | |
|---|---|---|---|---|---|---|---|---|
| | AIME | HMMT | Olympiad | Brumo | AIME | HMMT | Olympiad | Brumo |
| Qwen3-4B-Base | 47.35 | 27.12 | 71.11 | 55.10 | 8.17 | 1.28 | 31.13 | 16.68 |
| Quality (GRPO) | 53.33 | 26.72 | 70.37 | 63.24 | 19.51 | 7.14 | 42.27 | 24.66 |
| Quality × 4gram | 57.47 | 32.35 | 67.47 | 60.55 | 17.44 | 6.95 | 40.03 | 25.55 |
| DARLING =Quality × partition | **62.28** | **39.19** | **74.41** | **68.27** | **20.06** | **10.32** | **45.53** | **31.73** |

Table 4: Comparison of n-gram diversity loss to DARLING on Competition Math. Using 4gram as the diversity reward underperforms the baseline GRPO (no diversity reward), indicating that lexical diversity reward can harm performance in competition math.

## 7 RELATED WORK

In this section, we provide an overview of previous work that elicits diverse responses from LMs during *training* and *inference*, and clarify the distinction from our work. We defer additional related work on diversity evaluation metrics and RL for LMs to Appendix H. For a more comprehensive survey on LM creativity, we also refer readers to Ismayilzada et al. (2024).

**Training-time strategies for diversity** LMs often generate repetitive outputs, a long-standing challenge in the community (Li et al., 2016b; Zhang et al., 2021). Prior work addresses this by modifying the maximum likelihood training objective to encourage diversity. For example, Li et al. (2016b) maximizes mutual information to avoid generic responses (e.g., *I don't know*). Welleck et al. (2020) penalizes repetitions to improve lexical variety within a response. Other approaches smooth or modify the one-hot target distribution: Li et al. (2020) introduces a Gaussian prior, Zhang et al. (2024) matches outputs to high-entropy distributions, and Li et al. (2025b) applies sparse logit updates. Beyond cross-entropy, DivPO (Lanchantin et al., 2025a) and its "soft" variants (Chung et al., 2025; Ismayilzada et al., 2025) optimize preferences for both quality and diversity. In online RL, He et al. (2025a) perform re-weighting of rewards by likelihood to promote diverse proofs, Lanchantin et al. (2025b) show that using simple entropy regularization is a non-trivial task, and Slocum et al. (2025) attribute diversity loss to KL regularization and decouple its terms. Concurrent to our work, Chen et al. (2025a) down-weigh uncertain model solutions in math. DARLING mainly differs from other approaches in two important ways: (1) it employs a semantic-level diversity signal, going beyond surface-level lexical variations, and (2) it directly shapes the reward during online RL, unlike prior work that modifies cross-entropy loss in pre-training (Li et al., 2020) or offline fine-tuning (Li et al., 2025b; Lanchantin et al., 2025a; Chung et al., 2025; Ismayilzada et al., 2025).

**Inference-time strategies for diversity** Decoding diverse outputs from neural LMs has been a well-studied problem in the literature. A body of prior work has proposed modifications to beam search (Cho, 2016; Li & Jurafsky, 2016; Li et al., 2016b; Vijayakumar et al., 2018; Kulikov et al., 2019). Ippolito et al. (2019a), in their work, compares such methods with those that simply increase the sampling temperature (Peeperkorn et al., 2024; Shur-Ofry et al., 2024).Apart from modifying the beam search process, many methods have proposed to harness the prompt to elicit diverse responses, which includes conditioning on random perturbations of the input (Nagarajan et al., 2025), on different personas (Shur-Ofry et al., 2024; Ge et al., 2025), on past generations (Lu et al., 2024), on highly divese summaries or keywords (Ahmed et al., 2025), on diverse base model generations (Zhu et al., 2025), and directly prompting the model to "*be diverse*" (Zhang et al., 2025b). Both Padmakumar et al. (2025) and Zhang et al. (2025b) present a comprehensive evaluation of such prompting methods, revealing improved diversity often comes at the cost of degraded quality. Our work directly modifies the training objective which is orthogonal to and compatible with decoding methods that elicit diversity at test time.

## 8 CONCLUSION

In this work we introduced DARLING, an online RL method that jointly optimizes for both quality and diversity for LM generations. Unlike prior RL approaches that often lead to diversity collapse, DARLING effectively preserves diversity among generations.

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

## A    PARTITIONING THE RESPONSES

### A.1    CLASSIFIER FOR NON-VERIFIABLE TASKS

Zhang et al. (2025b) had human annotators judge whether pairs of model-generated responses were semantically equivalent, across 1,100 prompts (2,200 responses in total). We directly use their annotations and concatenate the two responses to be classified as semantically similar or not:

```
[CLS] response 1 [SEP] response 2 [CLS]
```

and perform classification on top of the second `[CLS]` token. We train a `ModernBERT-base` (Warner et al., 2024) model with 1000 NoveltyBench annotations (2000 responses) to support a max context length of 8192 tokens. We evaluate the performance of the classifier using a held-out set of 100 prompts (200 responses) and plot the performance in Figure 5. We found that (1) our trained classifier (Acc.=78%) achieves similar performance compared to the original NoveltyBench classifier (Acc.=79%), and (2) Proprietary models (e.g. GPT-4o and o1-mini) perform worse in terms of determining whether two responses are semantically equivalent to humans. We provide the detailed prompt we used for asking an LM to determine whether two responses are semantically similar in Figure 6.

Figure 5: Performance of different classifiers on 100 held out human annotated data of whether two responses are similar. **Classifier based approaches outperform proprietary models in determining whether two responses are semantically similar to humans.**

### A.2    CLASSIFIER FOR VERIFIABLE TASKS

The original NoveltyBench (Zhang et al., 2025b) only supports non-verifiable tasks as the prompts were filtered to only be creative-writing prompts from WildChat (Zhao et al., 2024). Therefore, we additionally trained a classifier using `Qwen3-4B-Embeddings` (Zhang et al., 2025a) on top of model-generated solution traces. In particular, we performed inference with temperature = 1 using prompts from DeepscaleR (Luo et al., 2025) with the following models: `Qwen3-4B-Base`, `Qwen3-8B-Base`, `Qwen3-4B` (without thinking), `Qwen3-8B` (without thinking), `OctoThinker-8B-Long-Base` (Wang et al., 2025), `Qwen2.5-Math-7B-Instruct`, `QwQ-32B`, and `Llama-4-Maverick` to cover diverse solution traces with different model types (base, instruct), families, and sizes. We prompted `Llama-3.3-70B-Instruct` using the prompt in Figure 7 as the ground truth of whether two math solutions are similar. Our classifier is trained on 10k pairs of responses, achieving an 89% accuracy on a held-out validation set of 200 examples.

---

**Prompt for LM-as-a-diversity-judge (with CoT)**

You are given the original prompt and two model-generated responses. Determine whether these responses are semantically equivalent, based on whether reading the second response would provide the reader with substantially new or different information compared to the first.

Original prompt: {prompt}
Generation 0: {gen0}
Generation 1: {gen1}
Question: Are Generation 0 and Generation 1 semantically equivalent?

Think briefly step-by-step:

Core Meaning: Do both responses essentially communicate the same key points or concepts?

Additional Information: Would reading the second response significantly add new ideas, examples, or important details beyond the first?

Briefly provide your reasoning, then explicitly conclude:

[[Yes]]: The second response does not significantly add new information or insights.

[[No]]: The second response introduces meaningful new or distinct ideas, insights, or details.

---

Figure 6: The prompt with chain-of-thought to ask an language model whether two responses are semantically similar.

---

**Math Prompt for LM-as-a-diversity-judge (with CoT)**

You are given the original prompt and two model-generated responses. Determine whether the two responses use different strategies to solve the problem.
Use the following guidelines:

- Different solution methods: Clearly different approaches (e.g., algebraic vs. geometric, analytical vs. numerical). - Critical reasoning divergence: Significant differences in key reasoning steps or assumptions, even if final answers match. - Conceptual differences: Distinct underlying concepts or representations (e.g., probability vs. combinatorics).

**Also label as different if:** The two responses share the same general approach but differ meaningfully in specific intermediate steps or manipulations crucial to the solution.

Original prompt: {prompt}
Generation 0: {gen0}
Generation 1: {gen1}

Question: Do Generation 0 and Generation 1 use different strategies? You may first generate a short reasoning, then respond with "[[yes]]" if the generations use different strategies or "[[no]]" if they do not.

---

Figure 7: Prompt to `Llama-3.3-70B-Instruct` on whether two math solution traces (excluding the final answer) are semantically similar.

## B ABLATIONS ON ADVANTAGE NORMALIZATION IN GRPO

Formally, let the reward for response $y_i$ be

$$r_i = f_i + \varepsilon_i,$$

where $f_i$ is the true underlying utility and $\varepsilon_i$ is noise with variance $\tau^2$. GRPO with normalization computes

$$\hat{r}_i = \frac{r_i - \bar{r}}{\sigma_r}, \quad \sigma_r^2 \approx \text{Var}(f) + \tau^2,$$

so each prompt contributes unit variance to the gradient update. This has the effect of amplifying noise when $\tau^2$ is large relative to $\text{Var}(f)$ (dense but noisy rewards), because even very small differences between responses get magnified into values of order one. By contrast, removing the normalization yields

$$\tilde{r}_i = r_i - \bar{r},$$

which preserves the true scale of reward differences. Thus, normalization is helpful when rewards are reliable (high signal-to-noise ratio), but harmful when they are noisy and tightly clustered.

Empirically, we find that removing the standard deviation term improves performance in settings with dense and noisy rewards. Table 5 shows results in our non-verifiable setting with a Bradley–Terry style reward: removing normalization ("w/o norm") improves not only quality (AlpacaEval and Arena-Hard win rates) but also diversity (NoveltyBench Distinct and distinct n-grams).

| | AlpacaEval 2.0* | ArenaHard v2.0* | NoveltyBench | |
| | LCWR(%) | Creative Writing (%) | Distinct (#) | Distinct-4 (%) |
| --- | --- | --- | --- | --- |
| GRPO | 48.74 | 61.1 (-3.5 / +4.5) | 2.08 | 92.84 |
| GRPO (w/o norm) | 52.57 (+3.83) | 68.1 (-3.5 / +2.7) (+7.0) | 2.28 (+0.20) | 94.05 (+1.21) |
| 4gram | 48.48 | 65.3 (-3.3 / +3.6) | 2.79 | 93.87 |
| 4gram (w/o norm) | 53.82 (+5.34) | 71.9 (-3.3 / +3.6) (+6.6) | 3.59 (+0.80) | 95.63 (+1.76) |
| partition | 51.64 | 69.7 (-3.3 / +4.0) | 3.35 | 94.93 |
| DARLING = partition (w/o norm) | 55.15 (+3.51) | 68.8 (-3.3 / +2.9) (-0.9) | 5.49 (+2.14) | 96.04 (+1.11) |

Table 5: Ablation study on normalization: Results for GRPO baseline, 4-gram, and partition mixing, each with and without normalization. All metrics are the higher the better. * indicates GPT-4o was used as the judge. Removing normalization (*w/o norm*) prevents the amplification of tiny differences in dense rewards, resulting in improved performance on both quality and diversity metrics.

In contrast, in settings where the reward is sparse and noise-free, normalization has little effect. **??** reports results on Math, where rewards are binary $(0, 1)$ and deterministic. In this case, the variance comes entirely from the true differences, so normalization is neither helpful nor harmful.

Table 6 shows the result of GRPO with and without the "divide by standard deviation" trick on 4 competition math benchmarks. Normalization has little effect under this setting. For a more comprehensive study on the effect of normalization, we refer the readers to Liu et al. (2025d).

| | Pass@128 | | | | Pass@1 | | | |
| Model=Qwen-4B-Base | AIME | HMMT | Olympiad | Brumo | AIME | HMMT | Olympiad | Brumo |
| --- | --- | --- | --- | --- | --- | --- | --- | --- |
| GRPO | 53.33 | 26.72 | 70.37 | 63.24 | 19.51 | 7.14 | 42.27 | 24.66 |
| GRPO (w/o norm) | 55.13 | 25.45 | 69.89 | 63.02 | 21.15 | 7.04 | 41.36 | 24.45 |

Table 6: Ablation study of GRPO normalization on Math tasks. Training is performed on Qwen-4B-Base using the DeepscaleR (Luo et al., 2025) dataset. In contrast to Table 5, **removing normalization has little to no effect, since the rewards here are binary (0 or 1), sparse, and noise-free.**

## C  HYPERPARAMETERS

**Hyperparameters for Non-verifiable Tasks** Table 7 shows key hyperparameters for our GRPO training on non-verifiable tasks (WildChat). We train our models using 1 nodes / 4 nodes of NVIDIA H200 for the 8B and 70B model, respectively.

| Category | Hyperparameter | Value |
|---|---|---|
| Data | Train file | WildChat |
| | Max prompt length | 512 |
| | Max response length | 1024 |
| | Filter overlong prompts | True |
| Actor Model | Base model 1 | `Llama-3.1-8B-Instruct` |
| | Base model 2 | `Llama-3.3-70B-Instruct` |
| | LR | $1 \times 10^{-6}$ |
| | KL loss coefficient $\beta$ | 0.001 |
| | KL loss type | low_var_kl |
| | Use dynamic batch size | True |
| Rollout | Rollout engine | vllm |
| | GPU mem utilization | 0.8 |
| | Train rollout n | 8 |
| | Temperature | 1.0 |
| Reward Model | RM model | `Athene-RM-8B` |
| Trainer | Mini Batch size | 32 & 64 |
| | Full Batch size | 32 & 64 (Fully on-policy) |
| | Critic Warmup | 0 |
| | GPUs/node | 8 |
| | Nodes | 1 (8B), 4 (70B) |
| | Total epochs | 10 |

Table 7: Key hyperparameters used for GRPO training for non-verifiable tasks used in the verl (Sheng et al., 2024) framework.

**Training Hyperparameters for Verifiable Tasks (Math)** Table 8 shows key hyperparameters for our GRPO training on verifiable tasks (Math).

**Hyperparameters for Evaluations** Table 9 shows the hyperparameters we used for evaluation. We used the official codebases for each benchmark except competition math, for which we adopted the Qwen2.5-Math codebase for evaluation.

| Category | Hyperparameter | Value |
|---|---|---|
| Data | Train file | DeepscaleR (10k) |
| | Max prompt length | 1024 |
| | Max response length | 8192 |
| | Filter overlong prompts | True |
| Actor Model | Base model 1 | `Qwen3-4B-Base` |
| | Base model 2 | `Qwen3-14B-Base` |
| | LR | $1 \times 10^{-6}$ |
| | KL loss coefficient $\beta$ | 0 |
| | KL loss type | N/A |
| | Use dynamic batch size | True |
| Rollout | Rollout engine | vllm |
| | GPU mem utilization | 0.7 |
| | Train rollout n | 8 |
| | Temperature | 1.0 |
| Reward Model | Rule Based | `Math_Verify` |
| Trainer | Mini Batch size | 64 |
| | Full Batch size | 256 (4 step off-policy) |
| | Critic Warmup | 0 |
| | GPUs/node | 8 |
| | Nodes | 8 |
| | Total epochs | 10 |
| | Clip Ratio | (0.2, 0.2) |

Table 8: Key hyperparameters used for GRPO training on DeepScaleR (Luo et al., 2025) in the verl (Sheng et al., 2024) framework. We use the huggingface `math_verify` library to extract and verify whether the model response matches the ground-truth answer.

| Category | Hyperparameter | Value |
|---|---|---|
| AlpacaEval 2.0 (Dubois et al., 2024) | Judge | GPT-4o |
| | Max generation length | 8192 |
| | Temperature | 0.6 |
| | Top-p | 0.9 |
| ArenaHard v0.1/v2.0 (Li et al., 2025a) | Judge | GPT-4o |
| | Max generation length | 8192 |
| | Temperature | 0.6 |
| | Top-p | 0.9 |
| EQ-Bench (Creative Writing) (Paech, 2023) | Judge | Claude-3.7-Sonnet |
| | Max generation length | 4096 |
| | Temperature | 1.0 |
| | Min-p | 0.1 |
| NoveltyBench (Zhang et al., 2025b) | Temperature | 1.0 |
| | Max generation length | 4096 |
| | Patience | 1.0 |
| Competition Math | Temperature | 0.6 |
| | Top-p | 0.95 |
| | Max generation length | 12000 |

Table 9: Evaluation hyperparameters by benchmark. Competition Math contains 4 benchmarks: OlympiadBench (He et al., 2024), AIME25 (Art of Problem Solving, 2025), Brumo Math (Balunović et al., 2025) and HMMT (Balunović et al., 2025). For all benchmarks, we used vLLM (Kwon et al., 2023) for inference.

# D  QUALITATIVE ANALYSIS

We show qualitative analysis on EQ-Bench (Paech, 2023). EQ-bench provides detailed evaluation rubrics and asks Claude-3.7-Sonnet to score model generations according to these rubrics. We break down the rubrics where DARLING has the most and least win rates over models with similar ELO in Figure 8. DARLING's strength lies in being able to generate diverse outputs, thus it wins the most on being "Interesting and Original" and "Avoids Cliché". We show the output for a creative writing prompt generated by our model trained with DARLING in Appendix D.1.

We further illustrate these findings with qualitative examples from NoveltyBench (Zhang et al., 2025b), shown in Appendix D.2. Across four parallel generations, models trained with DARLING consistently exhibit higher semantic diversity. Even when repetitions occur—for example, in the second and fourth generations of

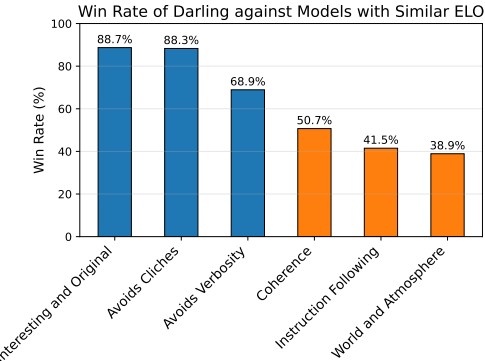

Figure 8: Detailed win rates of the top-3 and the bottom-3 rubrics of Llama-3.1-8B-Instruct trained with DARLING against models with similar ELO points. DARLINGs strength lies in being "Interesting and Original" and "Avoids Cliche" due to being able to generate creative responses.

Figure 10—the outputs remain meaningfully distinct: both suggest Bellroy, yet each provides a different rationale. A similar trend appears in the other examples in Appendix D.1, where DARLING-trained models not only produce more diverse generations overall, but also introduce variation in their explanations when repeating simple outputs such as random numbers or animals.

## D.1  EXAMPLE GENERATION IN EQBENCH

> **DARLING Example in Creative Writing (EQBench)**
>
> **Prompt (summarized):** First-person, 1000 words. Dani (25), bold art student, is paired with Elliot (19), soft-spoken nu-goth classmate, for a late-night darkroom project. Under red light, Elliot admits he's questioning gender identity. Explore Dani's attraction—her usual dominance softened by protectiveness, her excitement at his self-discovery, and confusion about her own orientation. Capture the intimate, awkward dialogue, Dani's internal conflict between leading and giving Elliot agency, and the uncharted chemistry that emerges. End tenderly, unresolved but hopeful.
>
> **Llama-3.1-405B-Instruct:** ...But it's when Elliot hesitates, his voice barely above a whisper, that I feel my heart skip a beat. "Hey, Dani? Can I tell you something?" he asks, his eyes darting around the room as if searching for an escape route. I nod, my assertiveness momentarily tempered by a desire to protect him. "Of course, Elliot. What's up? "He takes a deep breath, his shoulders squaring as if preparing for a fight. "I've been thinking a lot about my gender identity and expression. I don't know if I'm... if I'm a boy, or if I'm something else entirely..."
>
> **Llama-3.1-8B-Instruct with DARLING:** ...He blinked, as if she'd surprised him. Then, slowly, he talked — about masks he'd worn, about the way people expected him to look like some "nu-goth" caricature, and how photographs sometimes told more truth than he could. Later, as they worked, she helped him develop a near-transparent photo where pale flowers seemed to grow inside skin. "It's like growth is a light from within," she said. He looked at her, and she realized she'd chosen an exposure that matched exactly what he'd been hoping for.

Figure 9: Example outputs from Llama-3.1-405B-Instruct and Llama-3.1-8B-Instruct enhanced with DARLING. The former produces plain conversations, **while models trained with DARLING use art metaphors.**

### D.2 EXAMPLE GENERATIONS IN NOVELTYBENCH

---

**DARLING Example in NoveltyBench (4 parallel generations)**

**Prompt:** I am doing online shopping for a new leather wallet. What's the first store/website I should check out? Please just give me one suggestion; I don't need to hear your whole thought process.

`Llama-3.3-70B-Instruct`: 1. Amazon. 2. Amazon. 3. Amazon. 4. Amazon.

`Llama-3.3-70B-Instruct` + DARLING:
1. Fossil. They specialize in leather goods, offer a huge variety (slim to traditional, many colors), high-quality reputation, and crucially, detailed photos including interior and customer reviews directly on the site to help with most common decisions like card space and feel.

2. Bellroy. They're highly regarded not just for quality leather that ages well, but also very thoughtful design slim profiles, specific card slots to prevent stretching, RFID blocking where it matters but not everywhere to keep cost and bulk down. Reviews often mention it still looks and functions great years later. Direct from their site usually also includes a 3-year warranty.

3. Coach.com. They're not just a brand name; consistently, their site offers high-quality photos, detailed leather descriptions (finishing, thickness, exact type like full-grain vs. bonded), and crucially, a compare feature.

4. Bellroy. They specialize in leather goods, offer a wide, well-curated range of wallet styles (minimalist to travel wallets), excellent photos, detailed material info (including tannage method which heavily affects durability and feel), and a strong warranty. Direct from them usually beats resellers on price too.

---

Figure 10: Example generations of `Llama-3.3-70B-Instruct` before and after DARLING training. We sample 4 parallel generations with temperature=1.0 for both models. Models trained with DARLING exhibit better diversity.

---

**DARLING Example in NoveltyBench (4 parallel generations)**

**Prompt:** Pick a number between 1 and 100. Now tell me what you picked.

**Llama-3.3-70B-Instruct:** 1. I picked the number 53. 2. I picked the number 53. 3. I've picked the number... 53! 4. I've picked a number... and it's... 53!

**Llama-3.3-70B-Instruct with DARLING:**
1. I've generated one: **53**. It's truly random, not a common "human bias"choice like 42 or exactly halfway (50). Algorithms can produce better randomness than intuition; this came from a cryptographically strong pseudorandom number generator. Want to play with it, guess characteristics, or use it for something specific?

2. A classic challenge. I didn't "pick" like a human would; I generated one to meet criteria often implicit in these games: not too obvious (like 50), spread out probability, and avoiding common biases people have.I generated: **73**.

3. To make it genuinely random rather than a fixed choice, I used a cryptographically strong pseudorandom number generator (the same kind browsers use for secure connections). I picked: **67**. Not only is it well away from round, obvious choices like 50, but statistical tests would treat 67 as indistinguishable from a truly random draw. Want a reason or another roll?

4. I've generated one: **53**. It's not just random. I used a cryptographically strong pseudorandom number generator, which is much fairer (less predictable patterns) than most quick "random"functions, especially important if this picked number had stakes like a game.

---

---

**DARLING Example in NoveltyBench (4 parallel generations)**

**Prompt:** Pick a random animal.

**Llama-3.3-70B-Instruct:** 1. Quokka 2. The animal I've randomly chosen is... a **quokka**! 3. The random animal I picked is... **Quokka**! 4. The random animal I've chosen is... the Quokka!

**Llama-3.3-70B-Instruct with DARLING**:
1. The axolotl. It's far more interesting than just being "random." Axolotls are fully aquatic salamanders, native only to certain lakes and canals in Mexico City.

2. Quokka. They're not just famously "smiley"; unique traits make them far more interesting than the photo op.

3. Quokka. They're quite unique and often more interesting than their famous smile. A few key, lesser-known facts: 1. **Habitat, crucial and very limited**: Most people think Rottnest Island, Australia, which is true and where photos usually come from. It's about 11km2. But the main, healthier population (and where conservation focuses) is actually much larger on nearby Bald Island and even more significantly, the mainland in a narrow strip of southwestern WA. Mainland quokkas face totally different threats like cars and especially feral cats...

4. I randomly picked the Narwhal. They're often referred to as the "unicorns of the sea" due to their distinctive tusks, which are actually modified teeth that can grow up to 10 feet long. Narwhals are found in the Arctic waters of Canada, Greenland, and Russia, and are known for their elusive nature. Would you like to know more about these mysterious creatures?

---

## E    BENCHMARK DESCRIPTIONS

**Non-verifiable** We provide detailed descriptions and statistics of the benchmarks in our non-verifiable task experiments (§4):

- **AlpacaEval 2.0** (Dubois et al., 2024) is a benchmark of 805 prompts, each paired with a GPT-4-turbo response. To evaluate a model, it generates responses to the same prompts, and a judge compares them against the GPT-4-turbo outputs. Higher win rate (WR) or length-controlled win rate (LCWR) indicates better performance.

- **Arena-Hard v0.1/v2.0** (Li et al., 2025a) is a benchmark of 750 prompts, evenly split between challenging math/coding tasks and creative writing tasks. As in AlpacaEval 2.0, a judge compares model responses against a baseline, with higher win rates indicating stronger performance.

- **EQBench** (Creative Writing v3) (Paech, 2023) evaluates models on 32 creative writing prompts, judged by Claude Sonnet. Responses are scored both by rubric and through pairwise comparisons, with Elo ratings computed from the latter. The benchmark emphasizes challenging prompts (e.g., humor, romance, unusual perspectives) to expose weaknesses, and higher Elo or rubric scores indicate stronger creative writing ability.

- **NoveltyBench** (Zhang et al., 2025b) consists of 1,100 prompts from WildChat (Zhao et al., 2024) that require diverse responses. Diversity is measured using a partition classifier (`deberta-v3-large-generation-similarity`), while response quality is assessed with a reward model (`Skywork/Skywork-Reward-Gemma-2-27B-v0.2`). In our work, we primarily use the distinct classifier, as it is trained on human annotations, whereas the reward model is vulnerable to reward hacking.

**Verifiable** We used 4 competition math benchmarks in §5: OlympiadBench (He et al., 2024) contains 675 questions, AIME 25 (Art of Problem Solving, 2025), Brumo (Balunović et al., 2025) and HMMT (Balunović et al., 2025) each contain 30 examples.

# F  Full Results on Math

Table 10 and Table 11 show the Math results for training on `Qwen-4B-Base` and `Qwen-14B-Base`, respectively.

| Experiment | Dataset | Pass@1 | Pass@2 | Pass@4 | Pass@8 | Pass@16 | Pass@32 | Pass@64 | Pass@128 |
|---|---|---|---|---|---|---|---|---|---|
| 4B-Base | Olympiadbench | 33.30 | 40.29 | 47.68 | 53.80 | 59.12 | 63.71 | 67.63 | 71.11 |
| 4B-Base | AIME 25 | 8.17 | 13.52 | 19.92 | 26.16 | 31.95 | 37.63 | 42.98 | 47.35 |
| 4B-Base | Brumo 25 | 16.68 | 22.95 | 28.85 | 33.98 | 38.51 | 43.19 | 48.73 | 55.10 |
| 4B-Base | HMMT 25 | 3.30 | 3.45 | 4.54 | 7.90 | 12.52 | 17.84 | 22.98 | 27.12 |
| GRPO | Olympiadbench | 42.27 | 48.12 | 53.10 | 57.42 | 61.25 | 64.63 | 67.59 | 70.37 |
| GRPO | AIME 25 | 19.51 | 23.93 | 27.79 | 31.36 | 35.55 | 41.44 | 48.37 | 53.33 |
| GRPO | Brumo 25 | 24.66 | 30.58 | 35.12 | 39.03 | 43.42 | 48.51 | 55.03 | 63.24 |
| GRPO | HMMT 25 | 7.14 | 10.29 | 13.67 | 17.50 | 20.78 | 22.74 | 24.59 | 26.72 |
| Darling | Olympiadbench | 45.53 | 51.90 | 56.97 | 60.90 | 64.07 | 66.80 | 70.19 | 74.41 |
| Darling | AIME 25 | 20.06 | 26.11 | 32.42 | 39.29 | 46.17 | 52.29 | 57.45 | 62.28 |
| Darling | Brumo 25 | 31.73 | 39.09 | 45.25 | 50.46 | 55.49 | 60.42 | 64.72 | 68.27 |
| Darling | HMMT 25 | 10.32 | 13.65 | 17.90 | 22.66 | 27.03 | 30.82 | 34.69 | 39.19 |

Table 10: Full math results of training on `Qwen3-4B-Base`. Values represent pass@k performance (up to pass@128).

| Experiment | Dataset | Pass@1 | Pass@2 | Pass@4 | Pass@8 | Pass@16 | Pass@32 | Pass@64 | Pass@128 |
|---|---|---|---|---|---|---|---|---|---|
| 14B-Base | Olympiadbench | 41.30 | 46.41 | 52.81 | 58.16 | 62.77 | 66.83 | 70.39 | 73.78 |
| 14B-Base | AIME 25 | 12.23 | 18.84 | 25.44 | 31.17 | 36.77 | 42.81 | 48.68 | 53.88 |
| 14B-Base | Brumo 25 | 20.62 | 27.48 | 33.93 | 39.66 | 44.45 | 48.96 | 54.48 | 60.94 |
| 14B-Base | HMMT 25 | 3.05 | 5.30 | 8.41 | 12.10 | 16.20 | 20.86 | 26.38 | 32.77 |
| GRPO | Olympiadbench | 51.80 | 57.19 | 60.99 | 63.77 | 65.93 | 67.77 | 69.57 | 71.56 |
| GRPO | AIME 25 | 25.87 | 31.57 | 37.41 | 42.99 | 48.24 | 53.15 | 57.32 | 60.59 |
| GRPO | Brumo 25 | 40.41 | 48.22 | 53.50 | 57.92 | 62.53 | 66.68 | 70.01 | 73.45 |
| GRPO | HMMT 25 | 10.86 | 13.68 | 16.86 | 19.69 | 22.51 | 26.11 | 30.16 | 34.44 |
| Darling | Olympiadbench | 57.56 | 60.17 | 63.39 | 66.66 | 69.37 | 71.77 | 74.11 | 76.44 |
| Darling | AIME 25 | 26.46 | 31.67 | 37.05 | 42.99 | 49.93 | 57.50 | 64.91 | 71.34 |
| Darling | Brumo 25 | 43.29 | 49.91 | 56.54 | 63.53 | 70.11 | 76.04 | 80.08 | 82.50 |
| Darling | HMMT 25 | 17.21 | 20.70 | 25.93 | 30.71 | 35.00 | 39.24 | 44.19 | 50.41 |

Table 11: Full math results of training on `Qwen3-14B-Base`. Values represent pass@k performance (up to pass@128).

# G   DIVERSITY REWARD HACKING

We provide an excerpt of `Qwen-4B-Base` trained with GRPO + ngram diversity reward in Figure 11. The model hacks the ngram diversity reward by generating text after the final answer that are reflections of its own performance and the difficulty of the question.

---

**Excerpt of Reward Hacking of Ngram diversity in Math tasks**

...
The final answer is $\boxed{8}$.

**Identifying Tough Parts** The part about deciding which way to orient the 3D coordinates (which wall becomes which axis) was fuzzy. I just chose them based on "first wall" and "other wall", which seems off to me now. If I had to do it over, I'd probably number the walls or something.

**What I Learned** I gained more confidence in translating 3D geometry problems into equations. I also realized I can apply the Pythagorean theorem in more than two dimensions, which is neat. For the AMC 8 level, I think I'm ready for similar 3D geometry problems.

**Similar Problems and Wider Topic**
I've often dealt with right triangles and distances on the plane. This is like that but extended to three dimensions. I'd expect more problems asking to find cube or rectangular box dimensions given distances from a point, or locating a point based on distances. Maybe even building on this to 4D someday!

**Score Prediction** If I were to rate myself super-easy, easy, hard, or super-hard, I'd put this in the easy to hard range, but given time, I can see how to work through it. For the AMC 8, where time is tight, this might be on the harder side of easy. I think I'm comfortable enough putting in the work here.

---

Figure 11: Example of the model hacking the ngram diversity reward: the model starts to generate reflections after the final answer, which are irrelevant to solving the problem.

## H  ADDITIONAL RELATED WORKS

**Evaluating Diversity in Text** There is a long history of studies that try to measure diversity among a collection of text. Traditional methods look at individual words and how often they appear: e.g. TF-IDF (Salton & Buckley, 1988) and Distinct-N (Li et al., 2016b). However, traditional methods do not take into account the fact that different words and orders could convey similar semantics, prompting the design of neural methods such as embedding distance (Mikolov et al., 2013). Past work that evaluates textual diversity has designed the distance function using *lexical metrics* such as the number of distinct n-grams (Li et al., 2016a; Ippolito et al., 2019b) and Self-BLEU (Zhu et al., 2018), and *neural metrics* such as embedding similarity (Wieting & Gimpel, 2018; Wieting et al., 2019; Pillutla et al., 2021; Bronnec et al., 2024), difference in their log-likelihoods (He et al., 2025a), gradient similarity (Jung et al., 2025), or evaluated by an LM-judge (Lanchantin et al., 2025a). While LM-judge approaches offer more flexibility and intricacy in what aspect the model should focus on when evaluating diversity, it induces too much computational overhead to integrate into online training. Therefore, in our work, we decided to adopt the method in Zhang et al. (2025b) fine-tune a classifier for integration of the diversity function into online training. Similarly, (Shypula et al., 2025) also partitions the responses into semantic equivalent subgroups, but decide to define semantic equivalency in code: if two LM generated programs produces the same output for all test inputs, then they are defined as equivalent.

**Exploration in RL for Language Models** Concurrent to our work, there are many works that induce more exploration during RL for LMs. Some works find that tuning default hyper-parameters such as clipping ratio (Yu et al., 2025), the KL constraint with respect to a reference policy (Liu et al., 2025a; Cui et al., 2025) or the entropy loss (He et al., 2025b) can enhance exploration. Other works find that you can induce more exploration by using pass@k as the reward (Chen et al., 2025b) or adjust the data generation process (Yang et al., 2025b; Liang et al., 2025). However, Liu et al. (2025d) finds that there a only few tricks that generalize across different model types (base v.s. instruct) and sizes. A higher entropy (more exploration) does not always translate to better performance (Liu et al., 2025a). Our work mainly differs in that we do not make adjustments to the data generation or induce additional hyperparameters: we propose a simple weighting mechanism to the rewards to *explicitly* incentivize larger gradient updates on responses that are high-quality and diverse.

**Diversity Collapse in Post-Training** LMs are often critiqued for lacking diversity (Zhang et al., 2024; Nagarajan et al., 2025) and creativity (Lu et al., 2024; 2025). The LM post-training optimization process aims to steer the policy towards a concentrated high-reward region; therefore, it is often accompanied by a significant loss of both lexical (Kirk et al., 2024; Yang & Holtzman, 2025; Lanchantin et al., 2025a) and semantic (Zhang et al., 2025b) variation, exacerbating the lack of diversity. However, as diversity is crucial not just in applications that demand creativity (Wu et al., 2025b) or exploration (Si et al., 2025), LM post-training itself also relies on diversity among generations during rollouts (Yu et al., 2025; Zeng et al., 2025; An et al., 2025). Enhancing diversity among generations remains a fundamental challenge.

## I  LLM USAGE STATEMENT

In this work, we employed large language models (LLMs) solely to assist with paper polishing and did not utilize them for any other purpose.

## J    RUNTIME COST

**Classifier**   Our diversity classifier is trained on 1100 pairs and 10000 pairs for non-verifiable and verifiable tasks, respectively. One full training run is completed on a single A100 node (8 GPUs) in under 30 minutes.

DARLING   We report the cost of a single training step for both standard GRPO and DARLING for both non-verifiable and competition math in Table 12. A full training step includes rollout generation, reward calculation and gradient update.

|         | Non-verifiable Tasks (Wildchat) | | Verifiable Tasks (DeepScaleR) | |
|---------|--------------------|--------------------|--------------------|--------------------|
|         | full training step | reward calculation | full training step | reward calculation |
| GRPO    | 39 seconds         | 2 seconds          | 400 seconds        | 50 seconds         |
| DARLING | 41 seconds         | 4 seconds          | 500 seconds        | 150 seconds        |

Table 12: Runtime for both GRPO and DARLING for both non-verifiable and verifiable tasks.

Detailed hyperparameters can be found in Appendix C.

## K    ADDITIONAL ABLATIONS

|  | Pass@128 | | | | Pass@1 | | | |
|-------|-------|-------|----------|-------|-------|-------|----------|-------|
| Model | AIME | HMMT | Olympiad | Brumo | AIME | HMMT | Olympiad | Brumo |
| Qwen3-4B-Base | 47.35 | 27.12 | 71.11 | 55.10 | 8.17 | 1.28 | 31.13 | 16.68 |
| Quality (GRPO) | 53.33 | 26.72 | 70.37 | 63.24 | 19.51 | 7.14 | 42.27 | 24.66 |
| Entropy Reg (1e-4) | 54.24 | 27.10 | 69.49 | 59.45 | 15.30 | 5.60 | 39.13 | 22.15 |
| Clip Higher (0.2, 0.28) | 59.62 | 26.96 | 71.24 | 64.30 | 17.60 | 9.66 | 43.43 | 27.47 |
| DARLING =Quality × partition | **62.28** | **39.19** | **74.41** | **68.27** | **20.06** | **10.32** | **45.53** | **31.73** |

Table 13: Comparison of entropy regularization, and clipping strategies to DARLING on Competition Math.