# OpenReview forum: "Jointly Reinforcing Diversity and Quality in Language Model Generations"
_ICLR.cc/2026/Conference — Submitted to ICLR 2026_

### Official Review · Reviewer_mutG · 2025-10-28

**Soundness:** 3
**Presentation:** 3
**Contribution:** 3
**Rating:** 6
**Confidence:** 4

**Summary:**

This work proposes a new reward function that simultaneously considers generation quality and diversity. This reward function will give high reward to responses who are of higher quality and different from other responses. In particular, when measuring the response diversity, this work focuses on semantic level and trains a (binary) classifier to check if two responses are of the same semantic meaning.

To validate their method, this work evaluates on two tasks: non-verifiable (e.g., creative writing) and verifiable (e.g., AIME 25). On non-verifiable tasks, this work demonstrates a better Pareto frontier of the trade-off between quality and diversity; on verifiable tasks, this work shows high pass@1 and pass@k rates.

**Strengths:**

1, I appreciate the comprehensive set of experiments, including different (non-verifiable and verifiable) tasks, model classes and sizes, and metrics. These experiments validate the generality of the proposed method.

2, The proposed method is sound and obtains strong empirical results. The ablation studies are also informative, explaining how different designs affect the performance.

3, It shows promising results on improving RL performance by generating diverse outputs. As discussed in the abstract, high diversity might contribute to more exploration during online RL training. Although this claim is not fully explored, it would be very interesting to empirically verify the hypothesis.

**Weaknesses:**

1, This method relies on a binary classifier to measure semantic similarities between response pairs. Therefore, the whole training pipeline will be significantly affected by the quality of the binary classifier, especially considering the reward multiplicative aggregation. If the binary classifier is suboptimal, even high quality responses might not obtain high reward.

2, Three methods (GRPO, DivPO, GRPO-Unlikeness) are selected as baselines in section 4.1. However, in most experiments, only base model and GRPO are considered. It would be good if other method results can be reported.

3, While the method is sound, its contribution on algorithm is limited.

Minor:
1, While Figure 1 explains the sampling and semantic clustering, it does not reflect the quality. It would be good to explain in Figure 1 on how quality affects the final advantage.

**Questions:**

1, In line 173 to 177, modifications to the GRPO objective function is discussed. However, in equation (6), it is not clear how the token-level averaging and standard deviation are incorporated. Can you explain more about how equation (6) is different from the original GRPO objective?

---

> ### Author Response · Authors · 2025-11-14
> **Author Response**
>
> We sincerely thank the reviewer for the discussion. We are grateful that the reviewer finds our experiments “comprehensive”, our method “sound”, and our results “promising”. We hope to address the reviewer’s’ concerns below:
>
> >  If the binary classifier is suboptimal, even high quality responses might not obtain high reward.
>
> **We did not heavily optimize our diversity classifier.** As shown in Appendix A, it achieves 78% accuracy on human-labeled non-verifiable tasks and 89% on Llama-3.3-70B-labeled math data, indicating that the diversity signal is far from perfect. Despite this, our method still delivers strong empirical gains. This suggests that our approach does not rely on a highly accurate classifier to be effective. We expect even larger improvements with a more carefully tuned or task-specific classifier.
>
>
> > Three methods (GRPO, DivPO, GRPO-Unlikeness) are selected as baselines in section 4.1. However, in most experiments, only base model and GRPO are considered. It would be good if other method results can be reported.
>
> We reported the full results for GRPO, DivPO and GRPO-Unlikeliness for our non-verifiable tasks in Table 1. In our initial experiments, we tried many configurations and found that RL training with GRPO-Unlikeliness on verifiable tasks is unstable. Therefore, we decided to exclude GRPO-Unlikeliness in reporting our verifiable task results.
>
> > Minor: 1, While Figure 1 explains the sampling and semantic clustering, it does not reflect the quality. It would be good to explain in Figure 1 on how quality affects the final advantage.
>
> To clarify, we added a sentence in the Caption in Figure 1 saying that “high quality responses get positive advantage and low quality responses get negative advantage”. (Highlighted in green).
>
> > 1, In line 173 to 177, modifications to the GRPO objective function is discussed. However, in equation (6), it is not clear how the token-level averaging and standard deviation are incorporated. Can you explain more about how equation (6) is different from the original GRPO objective?
>
> Our method (Equation 6) differs from the original GRPO in 3 ways:
>
> (1) We incorporated a diversity loss.
>
> (2) We changed the original sentence level averaging to a token level averaging.
>
> (3) We removed the normalization by standard deviation in Equation 3.
>
> To clarify, the “baseline GRPO” in our paper also performs (2): the token-level averaging, which is the default implementation in verl and has shown to perform better than baseline GRPO [1].  We performed an ablation of (3) in Appendix B.
>
> **We look forward to addressing any remaining questions during the rebuttal and hope these updates warrant a higher evaluation.**
>
> ### References
> [1] Understanding R1-Zero-Like Training: A Critical Perspective (Liu et al., 2025)

---

> > ### Comment · Reviewer_mutG · 2025-11-17
> >
> > I thank the authors for the informative responses. I am glad to see that the performance is robust against the binary classifier performance; and thank for the clarification on differences from original GRPO. These justifies this work as a meaningful contribution to the research community. On the other hand, I still think the algorithmic contribution is only modest. So, overall, I decide to keep my original score.

---

### Official Review · Reviewer_pmFg · 2025-10-28

**Soundness:** 4
**Presentation:** 4
**Contribution:** 2
**Rating:** 6
**Confidence:** 4

**Summary:**

This paper presents a framework that jointly optimizes both quality and semantic diversity and thus addresses a very important aspect . The main method, called DARLING, introduces a partition function that measures diversity in meaning rather than just surface-level lexical form. The online reinforcement learning encourages distinct and high quality responses. The paper has results in both verifiable and non-verifiable tasks. In verifiable rewards, the method has higher pass@1 and higher pass@k where the increased diversity improves online exploration. In addition to the optimization framework, the paper also introduces a scalable semantic classifier. The authors claim the contribution of showing that diversity promotes greater exploration which has been shown in previous work and is a very intuitive conclusion in itself.

**Strengths:**

1. The paper is well written, easy to follow and has clearly labeled figures

2. The works deals with a very imporant problem with LLMs

3. The results are significant

4. The qualitative examples are very helpful

5. The ablations are rigorous, especially the one on how to use the two reward scores of the two different objectives

**Weaknesses:**

1. The results lack error bars. They are important especiallty in places like NoveltyBench results in Table 1.

2. There are missing citations and analysis of the original Semantic Entropy paper [1] which is highly relevant to the metrics you use

3. Papers like [2], [3] and [4] are highly relevant and deal with the same problem.

4. The third listed contribution in the paper showing that diversity helps exploration has been made before in papers like [4] and alluded to in most semantic entropy papers.

5. In section H, the authors claim that an LLM-judge is not used since "it induces too much computational overhead to integrate
into online training". Training a classifier on top of RL training can be a lot of overhead too. I think the authors should clarify and support this pivotal argument with more evidence.

[1] Kuhn, Lorenz, Yarin Gal, and Sebastian Farquhar. "Semantic uncertainty: Linguistic invariances for uncertainty estimation in natural language generation." arXiv preprint arXiv:2302.09664 (2023).

[2] Zhang, Yiming, et al. "Forcing diffuse distributions out of language models." arXiv preprint arXiv:2404.10859 (2024).

[3] Zhu, Alan, et al. "BARE: Leveraging Base Language Models for Few-Shot Synthetic Data Generation." arXiv preprint arXiv:2502.01697 (2025).

[4] Ahmed, Eltayeb, et al. "Intent Factored Generation: Unleashing the Diversity in Your Language Model." arXiv preprint arXiv:2506.09659 (2025).

**Questions:**

1. Can you please clarify what x and y stand for in equation 5? I can guess but it should be more clearly written.

2. What is the overhead of training a classifier in addition to the RL training? The paper has ablations on the accuracy of the classification with and without training, but it is important to show how a few-shot prompted classifier influences the final results as that is the most general mode of use. Also refer to point 5 in Weaknesses.

With the weaknesses being addressed, the additional citations being added (or justified why they should not be added) and most importantly with the overhead cost analysis, I am happy to increase my score.

---

> ### Author Response · Authors · 2025-11-14
> **Author Response**
>
> We sincerely thank the reviewer for the insightful comments and would like to extend our heartfelt gratitude that the reviewer finds our paper “well written”, our results “significant” and our ablations “rigorous”. We hope to address concerns in our response below:
>
> > The results lack error bars. They are important especiallty in places like NoveltyBench results in Table 1.
>
> We have modified Figure 1 to include the error bars for NoveltyBench results across all temperatures based on 5 runs with different random seeds. For Math, we did account for the variance that we generated 256 examples for each question for a more accurate estimate of from pass@1 to pass@128.
>
> > There are missing citations and analysis of the original Semantic Entropy paper [1] which is highly relevant to the metrics you use
> > Papers like [2], [3] and [4] are highly relevant and deal with the same problem.
>
> We thank the reviewer for the reminder, and apologize for our oversight. We do agree this is relevant work and we made changes (highlighted in green) in section 3 and 7 to include [1, 3, 4] in our manuscript. We have already cited [2] on pages 1, 9 and 19.
>
> > The third listed contribution in the paper showing that diversity helps exploration has been made before in papers like [4] and alluded to in most semantic entropy papers.
>
> We thank the reviewer for the observation. Indeed, [4] demonstrates that diversity facilitates exploration in DPO and Rejection Fine-tuning (STaR). **Our distinct contribution is demonstrating this effect in online RL.** We show that doing so leads to better-quality policies in both verifiable and non-verifiable settings. We have revised section 1 to clarify this distinction and contribution (changes highlighted in green).
>
> > Training a classifier on top of RL training can be a lot of overhead too.
>
> We emphasize that both training and using the diversity classifier introduce **minimal additional cost** to RL training.
>
> **Training cost:** The classifier is fine-tuned on only 1.1k (non-verifiable) and 10k (verifiable) generation pairs, requiring less than 30 minutes on a single A100 node (8 GPUs).
>
> **Runtime cost:** During online RL, the diversity judge is efficiently served via vLLM with concurrent queries. In non-verifiable tasks, the average reward computation time increases from 2s/step (GRPO) to 4s/step (Darling)—negligible compared to the 40s/step rollout generation. For verifiable tasks with longer generations, the increase is from ~50s to ~150s/step, still minor relative to the ~400s/step rollout time.
>
> Overall, the added engineering and computational overhead is **minimal** compared to the total RL training cost. We added a “runtime cost” section of training the classifier, and cost for running Darling in Appendix J.
>
> > What is the overhead of training a classifier in addition to the RL training? The paper has ablations on the accuracy of the classification with and without training, but it is important to show how a few-shot prompted classifier influences the final results as that is the most general mode of use. Also refer to point 5 in Weaknesses.
>
> The training cost of the classifier is very minimal: we trained both embedding-based classifiers under 30 minutes on a single A100 node (8 GPUs). However, serving an generative classifier (an LM-judge) as the diversity classifier during online RL is very time-consuming. In our initial attempts, when using an LM-judge as the classifier, each training step can take more than 15 minutes, therefore we opted to use our trained embedding based classifier.
>
> > Can you please clarify what x and y stand for in equation 5? I can guess but it should be more clearly written.
>
> $x$ stands for the input (user prompt) and $y$ stands for the model response. We introduced such notations at the beginning of section 2. We revised our manuscript (highlighted in green) to remind readers what $x$ and $y$ stands for at equation 5.
>
> **We look forward to addressing any remaining questions during the rebuttal and hope these updates warrant a higher evaluation.**

---

> > ### Comment · Reviewer_pmFg · 2025-11-16
> >
> > Thank you for your clarifications and additions.
> >
> > My concerns have been addressed, I will increase my score accordingly.

---

### Official Review · Reviewer_bbMB · 2025-11-07

**Soundness:** 2
**Presentation:** 3
**Contribution:** 3
**Rating:** 4
**Confidence:** 2

**Summary:**

This paper propose Diversity-Aware Reinforcement Learning (DARLING) to address the diversity collapse issue of post-training. DARLING is an online RL framework that jointly optimizes for response quality and semantic diversity. DARLING partitions LLM generations into semantically equivalent clusters by using a learned binary classifier, and amplify the reward for high-quality responses that are diverse (measured by how many responses are not in the the target response's semantic cluster. DARLING improves diversity and quality on instruction-following and math tasks.

**Strengths:**

This paper presents a novel method to incorporate a learned diversity classifier to the reward signal for post-training. The use of response partitioning to turn the binary classifier output into a continuous diversity score is clever. They also report higher diversity and quality scores than baselines (GRPO without DARLING, DivPO).

**Weaknesses:**

The range of tasks studied to verify the effectiveness of the method is rather narrow. I would love to see comparisons to baselines on more tasks, such as the two other tasks from DivPO paper (persona generation, keyword generation), which would further convince me of how well DARLING works.

DARLING also does not seem easily scalable, the paper does not discuss the computational overhead of calculating the diversity scores for n rollouts, where the method must perform O(n^2) pairwise inferences with the binary classifier. Also, the learned classifier needs data which may prevent the method from scaling to other domains.

**Questions:**

How much does the success of DARLING depend on the accuracy of the semantic classifier?
What happens when all model rollouts are semantically equivalent?

---

> ### Author Response · Authors · 2025-11-14
> **Author Response**
>
> We thank the reviewers for the insightful comments, and would like to extend our appreciation that the reviewer finds our method “novel” and “clever”. We hope to address all concerns below:
>
> > I would love to see comparisons to baselines on more tasks, such as the two other tasks from DivPO paper (persona generation, keyword generation)
>
> We appreciate the suggestion to include the additional DivPO tasks. In our work, we chose to focus on settings we view as more realistic and impactful, namely instruction following on WildChat data and verifiable mathematical reasoning. As reviewer `mutG` noted, our evaluation is already “comprehensive”, covering two categories (non-verifiable and verifiable math) across **eight benchmarks** (AlpacaEval, ArenaHard, EQBench, NoveltyBench, and four competition-math benchmarks) and **four models** (Llama 8B/70B and Qwen-4B/14B), resulting in a total of **18** model-benchmark pairs. To the best of our knowledge, our work is the most comprehensive work on promoting diversity with online RL, as existing work either only evaluates on instruction following [1] or only on math [2], performing experiments on one single base model.
>
> We hope this provides sufficient evidence of the method’s generality. We would be very happy to discuss additional extensions if you believe there are scenarios not covered by our experimental evaluation, and we remain fully open to further suggestions during the discussion phase.
>
> > DARLING also does not seem easily scalable
>
> We acknowledge that DARLING introduces additional computational overhead, but such an overhead is minimal compared to the RL training cost.
>
> Theoretically, while the worst-case cost is $O(n^2)$, our use of a union–find structure helps avoid many redundant pairwise comparisons in practice.
>
> Empirically, the overhead is modest: using vLLM to serve the diversity judge allows concurrent queries, increasing the average reward-computation time per step from about 2 s (GRPO) to 4 s (DARLING), while rollout generation itself takes ~40 s. Even on longer, verifiable tasks, the added time (≈150 s vs. ≈400 s total) remains small compared to the overall runtime.
> We added a “runtime cost” section of training the classifier, and cost for running Darling in Appendix J (changes highlighted in green).
>
> > How much does the success of DARLING depend on the accuracy of the semantic classifier?
>
> **We did not heavily optimize our diversity classifier.** As shown in Appendix A, it achieves 78% accuracy on human-labeled non-verifiable tasks and 89% on Llama-3.3-70B-labeled math data, indicating that the diversity signal is far from perfect. Despite this, our method still delivers strong empirical gains. This suggests that our approach does not rely on a highly accurate classifier to be effective. We expect even larger improvements with a more carefully tuned or task-specific classifier.
> > Also, the learned classifier needs data which may prevent the method from scaling to other domains.
>
> While our semantic classifier indeed relies on data, we argue that this is a fundamental requirement for any method evaluating diversity. The notion of diversity is inherently domain-dependent [3]—what constitutes diversity in creative writing (e.g., stylistic variation) differs greatly from that in mathematical reasoning (e.g., problem-solving approaches). Developing a generalizable diversity classifier that applies across all domains remains an open challenge for future research.
>
> Furthermore, our training of the classifier was minimal. We trained on 1100 and 10,000 pairs of responses for non-verifiable tasks and verifiable tasks, respectively. Each training run requires less than 30 minutes on a single A100 node (8 GPUs), and the classifier accuracies are far from perfect. **Our method is effective even with limited data and a minimally trained classifier.**
>
> > What happens when all model rollouts are semantically equivalent?
>
> In practice, we found that cases where all generations are semantically equivalent are very rare. When all generations are equivalent, both standard GRPO and our method assign zero advantage uniformly.
>
> **We look forward to addressing any remaining questions during the rebuttal and hope these updates warrant a higher evaluation.**
>
> ### References
>
> [1] Diverse Preference Optimization (Lanchantin et al. 2025)
>
> [2] CDE: Curiosity-Driven Exploration for Efficient Reinforcement Learning in Large Language Models (Dai et al., 2025)
>
> [3] LLM Output Homogenization is Task Dependent (Jain et al. 2025)

---

> > ### Comment · Reviewer_bbMB · 2025-11-27
> >
> > Thank you for your clarifications! I will increase my score accordingly.

---

### Official Review · Reviewer_dZRz · 2025-11-11

**Soundness:** 2
**Presentation:** 3
**Contribution:** 2
**Rating:** 4
**Confidence:** 5

**Summary:**

The authors propose an LLMS RL post-training framework that jointly optimizes for quality and semantic diversity. It uses a "learned partition function" (a semantic classifier) to group similar responses, then fuses the quality reward with a diversity signal. This reward design can generate both high-quality and diverse responses (in less-frequent semantic groups), encouraging the model to find multiple distinct solution paths. It partially solved the diversity/entropy collapse issue in LLMs' RL post training.

**Strengths:**

- The proposed framework addresses the important trade-off issue between quality and diversity during RL training. From the experiment results, the proposed technique is effective across both verifiable tasks and non-verifiable tasks, improving pass@1 and pass@K jointly.

- The proposed semantic classifier measures diversity at the semantic level rather than the superficial lexical level, capturing a more genuine diversity measurement during training.

**Weaknesses:**

- The current experiment results are limited in establishing the proposed technique is advantageous. There are other ways to improve a model's entropy, for example changing the temperature to encourage more diverse and stochastic output, etc. The authors didn't compare empirically with other existing solutions to address the entropy collapse issue [1] - [6].

- The current design is more heuristic-based and the entire system's performance is highly dependent on the quality of this partition function. If the classifier groups answers poorly, it will provide a noisy or incorrect diversity signal, which could hurt the main quality training. The entire system's performance should not rely heavily on a external component that is trained separately.

- The introduced "learned partition function" acts as an extra classifier. This adds significant engineering complexity and computational overhead to the RL training loop. There is lack of discussion on this aspect.


[1] Dai et al. CDE: Curiosity-Driven Exploration for Efficient Reinforcement Learning in Large Language Models. 2025

[2] Li et al. Jointly Reinforcing Diversity and Quality in Language Model Generations. 2025

[3] Cui et al. The Entropy Mechanism of Reinforcement Learning for Reasoning Language Models. 2025

[4] Park et al. Clip-Low Increases Entropy and Clip-High Decreases Entropy in Reinforcement Learning of Large Language Models. 2025

[5] Shen, H. On Entropy Control in LLM-RL Algorithms. 2025

[6] Zhao et al. The Majority is not always right: RL training for solution aggregation. 2025

**Questions:**

See the above Weaknesses. Mainly including:

- Comparison with other diversity encouraging approaches

- Sensitivity of the designed classifier and justifying its design validity

RL training Computational complexity after introducing the learned partition function, as intuitively, it will add significant engineering complexity and computational overhead to the RL training loop. So, please justify the practicality of this design.

---

> ### Author Response · Authors · 2025-11-14
> **Author Response (1/2)**
>
> We sincerely thank the reviewer for their comments. We are glad that the reviewer finds the diversity problem we study “important“ and our classifier “capturing genuine diversity”. We address comments and questions below.
>
> > The authors didn't compare empirically with other existing solutions to address the entropy collapse issue [1] - [6].
>
>  Below we clarify how our contribution differs from and contributes to prior work:
>
> **Our method promotes semantic (sentence-level) diversity**: Most existing approaches—e.g., Clip-Cov [3], Clip-Higher [4], and temperature/entropy tuning—optimize **token**-**level entropy**. As the reviewer noted, our method instead targets **“genuine” semantic diversity at the sentence level**, which is crucial for long-form reasoning where diversity in meaning, not just tokens, matters.
>
> **We benchmarked against the most relevant sentence-level methods**: Among prior work, DivPO [7] and GRPO-Unlikeliness [8] are the closest because they also aim to increase diversity beyond individual tokens using RL. We include both in **Table 1**, and our method clearly outperforms them across all tasks. [1], [5] and [6] were released September 2025, which we view as concurrent work; none of them have public code available.
>
> **Token-level diversity fails on verifiable tasks**: We performed an ablation using a token-level proxy (unique n-grams; Section 6.2). While it helps on non-verifiable tasks, models exploit it on verifiable ones, generating novel but **irrelevant** content (examples in Appendix G). This highlights the need for **semantic**, not lexical, diversity.
>
> To further address the reviewer’s concern, we performed additional experiments on Clip-Higher [4] and Entropy-Regularization [5].
>
> | Model                                | AIME (P@128) | HMMT (P@128) | Olympiad (P@128) | Brumo (P@128) | AIME (P@1) | HMMT (P@1) | Olympiad (P@1) | Brumo (P@1) |
> |--------------------------------------|--------------|--------------|-------------------|----------------|-------------|-------------|-----------------|--------------|
> | `Qwen3-4B-Base`                      | 47.35        | 27.12        | 71.11            | 55.10         | 8.17        | 1.28        | 31.13          | 16.68       |
> | Quality (GRPO)                       | 53.33        | 26.72        | 70.37            | 63.24         | 19.51       | 7.14        | 42.27          | 24.66       |
> | Entropy Reg (1e-4)                   | 54.24        | 27.10        | 69.49            | 59.45         | 15.30       | 5.60        | 39.13          | 22.15       |
> | Clip Higher (0.2, 0.28)              | 59.62        | 26.96        | 71.24            | 64.30         | 17.60       | 9.66        | 43.43          | 27.47       |
> | **Quality × partition (method)**     | **62.28**    | **39.19**    | **74.41**        | **68.27**     | **20.06**   | **10.32**   | **45.53**      | **31.73**   |
>
> We used 1e-4 for the entropy coefficient and ratio (0.2, 0.28) for Clip-Higher experiments. As shown, both entropy regularization and clip-higher are unable to match the performance of semantic diversity (Darling). Our findings align with [9] that higher entropy does not always translate to better exploration.
>
> ### References:
> [1] Dai et al. CDE: Curiosity-Driven Exploration for Efficient Reinforcement Learning in Large Language Models. 2025
>
> [2] Li et al. Jointly Reinforcing Diversity and Quality in Language Model Generations. 2025
>
> [3] Cui et al. The Entropy Mechanism of Reinforcement Learning for Reasoning Language Models. 2025
>
> [4] Park et al. Clip-Low Increases Entropy and Clip-High Decreases Entropy in Reinforcement Learning of Large Language Models. 2025
>
> [5] Shen, H. On Entropy Control in LLM-RL Algorithms. 2025
>
> [6] Zhao et al. The Majority is not always right: RL training for solution aggregation. 2025
>
> [7] Lanchantin et al. Diverse Preference Optimization. 2025
>
> [8] He et al. Rewarding the Unlikely: Lifting GRPO Beyond Distribution Sharpening. 2025
>
> [9] Liu et al. Scaling Up RL: Unlocking Diverse Reasoning in LLMs via Prolonged Training. 2025

---

> > ### Author Response · Authors · 2025-11-14
> > **Author Response (2/2)**
> >
> > > If the classifier groups answers poorly, it will provide a noisy or incorrect diversity signal, which could hurt the main quality training.
> >
> > **We did not heavily optimize our diversity classifier.**  As shown in Appendix A, it achieves 78% accuracy on human-labeled non-verifiable tasks and 89% on Llama-3.3-70B-labeled math data, indicating that the diversity signal is far from perfect. Despite the noise, our method still delivers strong empirical gains. This suggests that our approach does not rely on a highly accurate classifier to be effective. We expect even larger improvements with a more carefully tuned or task-specific classifier.
> >
> > > The introduced "learned partition function" acts as an extra classifier. This adds significant engineering complexity and computational overhead to the RL training loop.
> >
> > Training and using the diversity classifier introduce **minimal additional cost** to RL training. To see this, here we provide specifics:
> >
> > **Training cost:** The classifier is fine-tuned on only 1.1k (non-verifiable) and 10k (verifiable) generation pairs, requiring **less than 30 minutes** on a single A100 node (8 GPUs).
> >
> > **Runtime cost:** During online RL, the diversity judge is efficiently served via vLLM batching for concurrent queries. In non-verifiable tasks, the average reward computation time increases from 2s/step (GRPO) to 4s/step (Darling), which is **negligible compared to the 40s/step rollout generation**. For verifiable tasks with longer generations, the increase is from ~50s to ~150s/step, still **minor relative to the ~400s/step rollout time**.
> >
> > Overall, the added engineering and computational overhead is **minimal** compared to the total RL training cost. We have added a “runtime cost” section of training the classifier, and cost for running Darling in Appendix J.
> >
> > **We look forward to addressing any remaining questions during the rebuttal and hope these updates warrant a higher evaluation.**

---

### Author Response · Authors · 2025-11-29
**Official Comment by Authors**

We sincerely thank the reviewers for their feedback. To facilitate the AC's decision, we provide a summary of our core contributions, our clarifications regarding major reviewer questions, and the new results we added during the rebuttal.

## Core Contributions

**Novelty:** We introduce DARLING, the first online LLM RL framework that optimizes semantic (sentence-level) diversity rather than token-level entropy, using a lightweight learned diversity classifier to encourage distinct reasoning paths.

**Scale & Rigor:** We conduct the most comprehensive evaluation of diversity-aware online RL for LLMs to date, spanning 8 benchmarks, 4 models, and both verifiable and non-verifiable settings (18 total model–benchmark pairs).

**Efficiency:** Our diversity classifier trains in under 30 minutes on a single A100 node, adds minimal runtime overhead relative to rollout generation, and is effective even with imperfect accuracy, making DARLING practical and scalable.

**Significance:** Diversity has emerged as a key driver in advancing the capability frontier of AI, as reflected by multiple recent best-paper awards at ICML [1] and NeurIPS [2] recognizing work that measures genuine, meaning-level diversity rather than superficial token variation. By incorporating semantic diversity signals directly into online RL, DARLING enables richer exploration and yields more capable reasoning policies.

## Reviewer Main Concerns & Our Responses

**Efficiency of training and deploying the diversity classifier** (`dZRz`,` bbMB`,` pmFg`):
We provided detailed measurements showing the classifier adds minimal overhead: training takes <30 minutes on a single A100, and inference via vLLM increases reward computation only slightly (2→4 seconds for non-verifiable tasks; 50→150 seconds for math), while rollout generation dominates total cost (40/400 seconds). Thus, DARLING remains practical and scalable in online RL.

**Sensitivity to imperfect classifiers** (`dZRz`,`bbMB`,`mutG`):
 We showed DARLING is robust to noisy semantic labels. Classifier accuracies of 78% (non-verifiable) and 89% (math) still yield strong, consistent improvements. This confirms DARLING does not depend on highly accurate classifiers and performs well even with limited or imperfect data.

## Rebuttal Additions & New Results
We strengthened the paper during the rebuttal (all changes highlighted in the revised version in green):

- Detailed runtime and scalability analysis (Appendix J): We now report: classifier training cost (<30 minutes), per-step reward-computation times (2→4s for non-verifiable; 50→150s for verifiable), and rollout times (40–400 s), demonstrating that DARLING’s overhead is modest relative to RL rollouts.

-  New baselines: Entropy regularization & Clip-Higher: We added systematic comparisons showing these entropy-based methods do not match DARLING’s joint gains in diversity and quality.

- Error bars for NoveltyBench: We updated Figure 1 to include error bars across five seeds for NoveltyBench.

- Expanded related work and positioning: We added and discussed Semantic Entropy and other closely related works, clarifying the relationship and differences.

- Clarifications of technical details: We clarified the roles of $x$ and $y$ in Eq. (5).

## Reviewer Recognition
We appreciate the reviewers’ recognition of the paper’s strengths, including:

“Comprehensive set of experiments… validate the generality of the method.”

“The method is sound and obtains strong empirical results.”

“The results are significant.”

“The qualitative examples are very helpful.”

“The ablations are rigorous, especially on how to use the two reward scores.”

“Promising results on improving RL performance by generating diverse outputs.”

## Final Comment
We believe our contributions:
 (1) a principled semantic-diversity reward, (2) an efficient and scalable diversity classifier, (3) comprehensive empirical validation and ablations, and (4) new evidence that diversity improves exploration in online RL, represent a meaningful step forward for encouraging diversity in post-training of LLMs.

We hope the AC finds the strengthened submission suitable for acceptance.

---
## References
[1] Roll the dice & look before you leap: Going beyond the creative limits of next-token prediction. Nagarajan et al., ICML 2025.

[2] Artificial Hivemind: The Open-Ended Homogeneity of Language Models (and Beyond). Jiang et al., NeurIPS 2025.

---

### Meta-Review · Area_Chair_s77P · 2026-01-06

**Summary:**

Overall this is a decent piece of work, with initially mixed scores on both sides of the boundary. Post-rebuttal, some scores increased, while some were maintained, with overall leaning positive slightly.

More importantly, beyond quantitative scores, a few minor niggling concerns remain. Although some reviewers found the results to be strong and comprehensive, overall the algorithmic contribution is modest, while the novelty is also modest, and the big-picture significance/impact is unclear. While the rebuttals addressed most major technical concerns, most reviewers ultimately were not particularly enthusiastic about the work.

Unfortunately, for a highly-selective venue like ICLR, while this work is generally solid and without major concerns, in comparison to other stronger, better-received submissions, this work not meet the high bar this time.

**Reviewer Concerns:**

(Pls see above)

**Reviewer Scores:**

-- mutG was explicit about maintaining score of 6.
-- pmFg was explicit about raising score to 8.
-- bbMB was explicit about raising score to 6.
-- dZRz: I hazard would maintain their score, with a small chance of increasing.

---

### Decision · Program_Chairs · 2026-01-26

Reject